# Integrated Transcriptomic and Metabolomic Analysis Reveal the Underlying Mechanism of Anthocyanin Biosynthesis in *Toona sinensis* Leaves

**DOI:** 10.3390/ijms242015459

**Published:** 2023-10-23

**Authors:** Jing Xu, Yanru Fan, Xiaojiao Han, Huanhuan Pan, Jianhua Dai, Yi Wei, Renying Zhuo, Jun Liu

**Affiliations:** 1State Key Laboratory of Tree Genetics and Breeding, Chinese Academy of Forestry, Beijing 100091, China; 2Key Laboratory of Tree Breeding of Zhejiang Province, Research Institute of Subtropical Forestry, Chinese Academy of Forestry, Hangzhou 311400, China

**Keywords:** *Toona sinensis*, transcriptome, metabolome, anthocyanin, transcription factor, plant hormones

## Abstract

*Toona sinensis*, commonly known as Chinese Toon, is a plant species that possesses noteworthy value as a tree and vegetable. Its tender young buds exhibit a diverse range of colors, primarily determined by the presence and composition of anthocyanins and flavonoids. However, the underlying mechanisms of anthocyanin biosynthesis in *Toona sinensis* have been rarely reported. To explore the related genes and metabolites associated with composition of leaf color, we conducted an analysis of the transcriptome and metabolome of five distinct *Toona* clones. The results showed that differentially expressed genes and metabolites involved in anthocyanin biosynthesis pathway were mainly enriched. A conjoint analysis of transcripts and metabolites was carried out in JFC (red) and LFC (green), resulting in the identification of 510 genes and 23 anthocyanin-related metabolites with a positive correlation coefficient greater than 0.8. Among these genes and metabolites, 23 transcription factors and phytohormone-related genes showed strong coefficients with 13 anthocyanin derivates, which mainly belonged to the stable types of delphinidin, cyanidin, peonidin. The core derivative was found to be Cyanidin-3-O-arabinoside, which was present in JFC at 520.93 times the abundance compared to LFC. Additionally, the regulatory network and relative expression levels of genes revealed that the structural genes *DFR*, *ANS*, and *UFGT1* might be directly or indirectly regulated by the transcription factors SOC1 (MADS-box), CPC (MYB), and bHLH162 (bHLH) to control the accumulation of anthocyanin. The expression of these genes was significantly higher in red clones compared to green clones. Furthermore, RNA-seq results accurately reflected the true expression levels of genes. Overall, this study provides a foundation for future research aimed at manipulating anthocyanin biosynthesis to improve plant coloration or to derive human health benefits.

## 1. Introduction

As the national economy grows rapidly, and living standards continue to improve, consumers today demand more than just food and clothing. They desire a healthy diet that also helps prevent diseases. Food that is high in nutrition and has antioxidant properties is particularly appealing [1]. Chinese Toon (*Toona sinensis*), a tree with both medicinal and edible uses, contains rich chemical substances with functions such as antifungal, antiglycation, or anti-tumor activities. Flavonoid compounds, especially anthocyanin, have been identified as the major bioactive constituents in *Toona* [2,3,4] which are considered to be of the most beneficial plant-derived antioxidants and have led to increased efforts to analyze the genetic controls of the mechanism of flavonoids synthesis [5,6].

Anthocyanins belong to the flavonoid class of phenolic compounds. Anthocyanins are water-soluble pigments present in plants that contribute to the colors of leaves, flowers, and fruit, which are also crucial for plant growth and beneficial for human health due to their ability to scavenge active oxygens [7]. More than 600 distinct anthocyanin types have been identified so far through previous research [8]. The most prevalent anthocyanin pigments, which generate purple, blue, and red colors, are cyanidin (Cy), delphinidin (Dp), pelargonidin (Pg), peonidin (Pn), petunidin (Pt), and malvidin (Mv) [9]. Anthocyanin biosynthesis and regulation is a complex process influenced by various internal and external factors [10,11].

The anthocyanin biosynthesis pathway is a well-characterized process in plants such as *Arabidopsis thaliana*, apple, pear and tomato [12,13,14,15,16,17,18]. The pathway is initiated by the phenylpropanoid and flavonoid pathways, which are controlled by both structural and regulatory genes. The expression of functional structural genes was related with the accumulation of anthocyanin [19]. Overexpression of enzymes such as dihydroflavonol-4-reductase (DRF), anthocyanidin synthase (ANS) and UDP-glucoside: flavonoid glucosyltransferase (UFGT) can increase color accumulation in fruits. Meanwhile, The MBW complex, which is formed by MYB-bHLH-WD40 transcription factors, regulates anthocyanin biosynthesis by promoting or inhibiting the expression of structural genes [18,20,21,22,23,24,25]. The MYB transcription factor TRANSPARENT TESTA 2 (TT2) in *Arabidopsis thaliana* interacts with TT8 (bHLH42) and TTG1 (WD40) to regulated the expression of late biosynthetic genes, resulting in pigmentation of the seed coat [26]. In addition, the transcription factor, such as WRKY, NAC, bZIP, and MADS, have also been proven to play a role in regulating anthocyanin biosynthesis. *LONG HYPOCOTYL 5 (HY5)*, a bZIP transcription factor, directly participates in photosensitive morphology, UV-B resistance, and anthocyanin biosynthesis [27,28]. Studies have revealed its interaction with B-Box proteins (BBX) to control anthocyanin biosynthesis [29,30]. In addition, plant hormones also interact with transcription factors to influence the regulation of anthocyanin accumulation [31,32]. Jasmonate ZIM-domain (JAZ) proteins interact with essential components of WD-repeat/bHLH/MYB transcriptional complexes, namely bHLH (Transparent Testa8, Glabra3 [GL3], and Enhancer of Glabra3 [EGL3]) and R2R3 MYB transcription factors (MYB75 and Glabra1), to repress JA (Jasmonic acid)-regulated anthocyanin accumulation and trichome initiation [33]. Identification of structural enzymes and regulatory factors is crucial for progress in understanding the underlying mechanism of anthocyanin biosynthesis regulation.

The investigation of anthocyanin-regulating genes and associated flavonoid metabolites is critical to elucidate the mechanism that governs the biological synthesis of anthocyanins. To study gene expression and metabolic changes in different cultivars, researchers mainly rely on transcriptomic and metabolic profiling, which complement each other [34,35]. In this study, we conducted a dual-level exploration of genes and metabolites that control the pigmentation of *Toona sinensis* leaves using transcriptome and metabolomics approaches. Specifically, we analyzed the transcriptome and metabolome data of five *Toona sinensis* clones exhibiting a range of leaf colors, including red, light red, pink, dark, and green, to clarify regulatory networks for biosynthesis-related genes and key metabolomes. Overall, our findings unveil the regulatory mechanisms driving the coloration of *Toona sinensis* leaves.

## 2. Results

### 2.1. Morphological Observation and Anthocyanin Content of Leaves in Toona sinensis

Five clones of *Toona sinensis* with distinct leaf colors were selected to analyze the regulatory mechanism of anthocyanin biosynthesis. Among the clones, JFC *Toona* is characterized by brilliant-red leaves and is commonly cultivated in Zhejiang province. The leaf color of other clones was pink buds (FHXN), light-red with a little green buds (BSH), dark-purple buds (HBG), and green buds (LFC) (Figure 1A). Then, the total anthocyanin was extracted and the resulting color were displayed in Figure 1B, which reflect the anthocyanin content of different clones. Total anthocyanin present in the leaves were quantified, with JFC *Toona sinensis* found to have the highest concentration of anthocyanin, followed by BSH, FHXN and HBG, and LFC (Figure 1C). Obviously, LFC exhibited green leaves due to a lack of anthocyanin accumulation.

### 2.2. Metabolome Analysis of Anthocyanin in Five Toona Clones

To compare the flavonoids and anthocyanin compound composition among five *Toona* clones, three biological replicates of each clone were sampled to detect anthocyanin-related metabolites employing the UPLC-MS/MS platform. The resulting metabolite composition data sets underwent PCA analysis (Figure 2A). The PCA plot for the anthocyanin-related metabolites showed a clear separation between the red, pink, light red, green, dark-purple samples. A total of 43, 40, 44, 40, and 32 anthocyanin-related metabolites were identified in leaves of JFC, BSH, FHXN, HBG, and LFC, respectively (Appendix A). A total of 31 anthocyanins were identified in all of the samples. Then, the |log2(fold-change)| ≥ 1 and *p* value < 0.05 was used to determine significantly metabolites between groups (Figure 2B). In group LFC vs. JFC, 23 upregulated metabolites and 8 downregulated metabolites were found to be significantly different (Figure 2B). In LFC vs. BSH, there were 24 upregulated metabolites and 6 downregulated metabolites (Figure 2B). In BSH vs. JFC, there were 10 upregulated metabolites and 11 downregulated metabolites (Figure 2B). Counting the differential metabolites compared with both LFC and JFC for each individual plant, a Venn diagram was used to show that there were 6 differential metabolites in the comparison group compared with JFC and 24 common differential metabolites compared with LFC (Figure 2C,D). Thirty-one common detected metabolites were used to draw clustering heatmap by TBtools (https://bio.tools/tbtools, accessed on 13 September 2023). The contents of cyanidin and delphinidin could be mainly responsible for the leaves color of *Toona sinensis* (Figure 2E).

Metabolomic analyses revealed significantly higher levels of several anthocyanin pigments, namely Naringenin, Delphinidin-3-O-rutinoside, Delphinidin-3-O-5-O-(6-O-coumaroyl)-diglucoside, Cyanidin-3-O-arabinoside, Cyanidin-3-O-glucoside, Cyanidin-3-O-rutinoside, Cyanidin-3-O-galactoside, Cyanadin-3-O-xyloside, Cyanidin-3-O-sambubioside, Cyanidin-3,5-O-diglucoside, Peonidin-3-O-rutinoside, in the JFC variety compared to other varieties. Interesting, among these anthocyanins, the content of Cyanidin-3-O-arabinoside was found to be 520.93-fold higher in JFC than in LFC.

### 2.3. Transcriptome Analysis of Five Toona sinensis Clones

Fifteen cDNA libraries (each clone with three biological replicates) were sequenced using the Illumina HiSeq platform. After removing low-quality reads, each sample yielded between 42.16 M to 52.03 M clean reads (Appendix A). The Q30 base exceeded 92.56% and the guanine–cytosine (GC) content of each sample was greater than 42.27%, indicating acceptable sequencing quality. After filtration, the clean data reached 6.32–7.80 Gb for each sample. The genome of *Toona sinensis* var. Heiyouchun was utilized, which marked a significant was the first chromosome-level genome assembled of Toona, as our reference genome [36]. Furthermore, comparison of all clean reads with this reference genome and subsequently performed gene annotations, the results revealed a high rate of similarity, ranging from 92.26% to 94.03%. These results validate the reliability of our data. Novel genes were defined as unigenes found in the sequencing results but not included in the reference genome (or reference gene set) after reconstructing the transcripts using StringTie v1.3.4d software. Comparison identified 9127 novel genes. The PCA results of transcriptome showed significant differences among five clones (Figure 3A).

To investigate the function of unigenes in *Toona*, we conducted a comparative analysis with various databases including KEGG, NR, Swiss-Prot, GO, KOG, and Trembl using BLAST (Diamond version 2.0.9) software. To obtain annotation information on unigenes, amino acid sequences were derived from the unigene sequences and compared against the Pfam database using HMMER V3.2 (http://hmmer.org/, accessed on 10 June 2022) software. Interestingly, when compared to data in the NR database, it was observed that gene sequences from *Toona sinensis* exhibited a greater degree in similarity to *Citrus clementina* (30.91%), followed by *Citrus sinensis* (30.47%) as depicted in Appendix A. The function terms of unigenes were annotated in the KOG and GO database, and the function groups were showed in Appendix A.

Based on the reference transcriptome, 40,421 genes were assembled from clean reads, which included 31,647 genes annotation in reference genome and 9127 novel transcripts. In addition, 23,909, 16,532, 25,617, 15,566, 32,348, and 25,575 unigenes were annotated by KEGG, NR, SWISSPORT, THEM, GO, and KOG database. There were 14,153 unigenes annotated by all of the databases. A comparison with six public databases (KEGG, NR, SwissProt, Tremble, GO and KOG) annotated a total of 13,295 differentially expressed genes (DEGs) and predicated their potential functions. The DEGs were counted with an absolute |log2fold-change| ≥ 1 and a false discovery rate (FDR) ≤ 0.05 in each compared groups. According to five clones of *Toona* leaves, 10 pairs of comparisons were performed and the number of up-regulated and down-regulated differential genes was counted in all groups (Figure 3B). Across all compared groups, a total of 310 DEGs were co-detected (Figure 3C). The KEGG-based enrichment analysis revealed that all DEGs compared to LFC were enriched into 140 KEGG pathways. These pathways included flavonoid biosynthesis, phenylpropanoid biosynthesis, and anthocyanin biosynthesis, all of which were significantly enriched (Figure 3D). These terms have previously been associated with anthocyanin production.

### 2.4. Analysis of Anthocyanidin Biosynthetic Pathway Genes and Anthocyanin Derivates in JFC, BSH, and LFC

According to transcriptome and metabolites studies, anthocyanin biosynthesis may be responsible for the distinct red leaf colorations of *Toona*. Therefore, it is necessary to investigate the mechanisms that underlie this process. Genes encoding anthocyanin biosynthetic enzymes were found to be associated with different compositions of the compounds in the anthocyanin biosynthesis pathways. We constructed a pathway diagram displaying the expression heat map of structural genes and compounds for red, light-red, and green leaves (Figure 4). The *PAL* gene catalyzed the conversion of L-Phenylalanine to cinnamic acid and was highly expressed in JFC. Among the *4CL* genes, two exhibited a gradual decrease in expression from red to green leaves across JFC, BSH, and LFC. CHS, the first key enzyme in the flavonoid pathway, played a pivotal role in catalyzing the synthesis of chalcone. Three *CHS* genes were differentially expressed in these clones. Two of these genes were significantly up-regulated in BSH and JFC, compared with LFC, whereas one was more highly expressed in LFC than in BSH and JFC. Two *CHI* genes responsible for converting naringenin chalcone to naringenin exhibited higher expression levels in BSH and lower expression in LFC. The accumulation of anthocyanin was positively correlated with the expression level of these genes. The study found that there was a gradual decrease in the content of naringenin metabolite in JFC, BSH, and LFC. Moreover, the study found that the *FLS*, *DFR*, and *ANS* genes play a crucial role in transforming naringenin into delphinidin, cyanidin, and pelargonidin, respectively, with their expression levels being up-regulated in JFC and down-regulated in LFC (Figure 4). UFGT is a critical enzyme involved in the biosynthesis of stable anthocyanins from unstable anthocyanin glycosides. The study identified two *UFGT* genes whose expression levels were significantly higher in JFC than in BSH, with the lowest levels observed in LFC, resulting in the highest accumulation of different anthocyanin compounds in JFC compared to LFC. These results revealed that the expression of genes in anthocyanin biosynthesis was consistent with accumulation of anthocyanins.

### 2.5. Correlation Analysis of Differentially Expressed Transcripts and Anthocyanin Compounds to Identify Key TFs and Plant Hormones Genes

To achieve the regulatory network between regulatory genes, including transcription factors and plant hormone-related genes, and anthocyanin metabolites, an association analysis was conducted between transcripts and metabolome (Appendix A). The major transcription factors and plant hormone-related genes were then identified using differentially expressed genes and metabolites that showed significant differences between JFC (red) and LFC (green). A conjoint analysis was conducted to establish relationships between 510 genes and 23 metabolites with a positive correlation coefficient greater than 0.8 within JFC and LFC. Among these genes and metabolites, 23 TFs and phytohormone-related genes were identified and showed strong correlations with 13 anthocyanin derivates (Table 1). Using Cytoscape 3.10.0, a regulatory network was constructed between these 23 genes and 13 anthocyanin derivatives (Figure 5A). The top 3 TFs associated with the biosynthesis of anthocyanins belonged to *AGL9* (MADS-box), *SOC1* (MADS-box), *MYB* (CPC) (Table 1). Furthermore, among the top five compounds, Cyanidin-3-O-arabinoside and Cyanidin-3-O-galactoside emerged as the most significantly different anthocyanin derivatives in *Toona sinensis* (Table 1). Consequently, a regulatory network was constructed between these top transcription factors and structural regulatory genes involved in anthocyanin biosynthesis (Figure 5B). These findings suggest that MADS-box, MYB, and bHLH transcription factors may regulate *DFR, ANS, UFGT,* and other structural genes involved in the biosynthesis of anthocyanin in *Toona sinensis* by either affecting or participating in the regulation of structural gene.

### 2.6. Expression Profiles of TFs and Plant Hormone Genes Related to Anthocyanin Biosynthesis

Previous studies have demonstrated that the biosynthesis of anthocyanins requires the involvement of transcription factors, such as MADS-box, MYB, bHLH, and bZIP. In this study, we identified a total of 11 MADS-box genes, 33 MYBs, 42 bHLHs, 36 bZIPs, and 15 WD40 genes in *Toona*. The RNA expression levels in JFC, BSH, and LFC was compared, which revealed that 12 *bHLH*, 10 *bZIP*, 12 *MYB*, 2 *MADS-box*, and 1 WD40 were significantly highly expressed in JFC, lowest in LFC (Figure 6A). Furthermore, the heatmap showed that all highly correlated TFs with anthocyanin derivates belonged to these groups.

Studies suggest that plant hormones have a crucial function in regulating anthocyanin concentrations in leaves and fruits via direct or indirect regulation of transcription factors responsible for anthocyanin biosynthesis. The hormones capable of activating or inhibiting the structural genes involved in the biosynthesis pathway include auxin, gibberellins, cytokinin, brassinosteroids, jasmonic acid, salicylic acid, and abscisic acid. Furthermore, we identified several differentially expressed genes that participate in plant hormone signal transduction, including 8 Auxin-related genes, 9 CTK-related genes, 10 GA-related genes, 23 BR-related genes, 7 ABA-related genes, 11 ethlye-responsive genes, 6 JA-related genes, and 3SA-related genes. Among these genes, the following exhibited a positive correlation coefficient greater than 0.8 with anthocyanin compounds: 2 Auxin-related genes (*Maker00008503* and *Maker00022256*), 1 CTK-related gene (*Maker00000350*), 1 GA-related gene (*Maker000117930*), 2 BR-related genes (*Maker00026791* and *novel.6183*), 1 Ethylene-responsive genes (*Marker00033233*), and 2 SA-related genes *(Maker00020104* and *Maker00032838*) (Figure 6B). This indicates that these genes may positively regulate anthocyanin biosynthesis either directly or indirectly.

### 2.7. Expression Analysis of Genes through qRT-PCR

To validate the RNA-seq results, we utilized qRT-PCR to evaluate the expression of 8 structural genes and 4 TFs involved in the anthocyanin biosynthesis pathway. Our observations revealed that *bHLH162*, *CPC*, and *SOC1* TFs were significantly upregulated in JFC and exhibited the lowest expression levels in LFC (Figure 7A). Additionally, there was a strong correlation between the RNA-seq and the characteristic leaf color of these samples. The qRT-PCR outcomes of structural genes were also significantly correlated with RNA-seq (Figure 7B). Moreover, our data suggested that the expression levels of *bHLH162* (*Maker00021798*, 2.99 times higher in JFC than in LFC), *CPC* (*Maker00014746*, 10.48 times), and *SOC1* (*Maker00029632*, 6.67 times) were synchronized with *DFR* (*Marker00011618*, 1.84 times), *ANS* (*Marker00000819*, 2.95 times), and *UFGT-1* (*Marker00029333*, 3.07 times), indicating that they may be involved in regulating anthocyanin production. These findings demonstrated that transcription data accurately reflect gene expression in the anthocyanin biosynthesis of *Toona sinesis*.

## 3. Discussion

Anthocyanins present in *Toona sinensis* leaves possess various health benefits, including antioxidant and anti-inflammatory properties, and potential cancer-preventing properties. Understanding the regulatory mechanisms involved in anthocyanin biosynthesis in *Toona sinensis* leaves has significant implications for human health and agriculture. In this study, we utilized metabolomic and transcriptomic approaches to elucidate the molecular mechanisms and regulatory network involved in anthocyanin biosynthesis in *Toona*. A total of 32–44 metabolites related to anthocyanins were identified in five different clones of *Toona* leaves using UPLC-MS/MS. We found that JFC had the highest total anthocyanin content, with Delphinidin, Cyanidin, Peonidin being the most abundant pigments in JFC. Specifically, Cyanidin-3-O-arabinoside was found to be 520.93 times more prevalent in JFC than in LFC, indicating that cyanidin compounds are the dominant anthocyanin in JFC. These findings are consistent with those of previous studies on strawberry petals, which have shown that cyanidins are the primary anthocyanin compound present [37].

Transcriptome sequencing of *Toona* leaves with different colors can provide valuable insights into the regulating genes involved in anthocyanin biosynthesis. The KEGG pathway enrichment analysis of differentially expressed genes (DEGs) in colored leaves of different clones compared to green leaves revealed significant enrichment of genes involved in flavonoid biosynthesis, phenylpropanoid biosynthesis, and the anthocyanin biosynthesis pathway. This finding supports previous studies demonstrating the involvement of multiple enzymes encoded by early biosynthesis genes and anthocyanin biosynthesis genes in anthocyanin biosynthesis. The study indicates that the biosynthesis of naringenin is a crucial step in the flavonoids metabolic pathway and plays a decisive role in the synthesis of anthocyanin compounds. The *PAL*, *C4H*, *4CL*, *CHS*, and *CHI* are key genes in the flavonoids metabolic pathway and play a decisive role in determining naringenin production. The study found that JFC had a higher naringenin content than BSH and LFC, but only two *4CL* genes (*Maker00007314*, *novel.5717*) were expressed at high levels in JFC and low levels in LFC. The expression of *4CL* genes was highly positively correlated with naringenin content, indicating that the number and function of genes in the naringenin pathway may affect naringenin content. During the biosynthesis of anthocyanins, DFR catalyzes the reduction reaction of flavonoid-3′,5′-hydroxylase to form anthocyanins, which was confirmed as a key step for regulating anthocyanin types. ANS converts anthocyanin precursors to anthocyanin glycosides. UFGT can catalyze the conversion of unstable anthocyanidins into stable anthocyanins. The absence of these genes directly affects anthocyanins biosynthesis, leading to pigment loss [38,39]. Down-regulation of the apple *DFR* gene can inhibit both cyanidin and procyanidin accumulation. In *Vitis vinifera*, the silencing of the DFR gene leads to the absence of anthocyanins [40]. In *Duchesnea indica*, a plant belonging to the Rosaceae family, decreased expression of the *ANS* gene results in white-colored fruits [41]. Conversely, overexpressing the *SmANS* gene in *Salvia miltiorrhiza* can increase anthocyanin content but decrease the biosynthesis of salvianolic acid [42]. In our study, we found significantly increased expression of the *DFR* gene (*Marker00011618*), *ANS* gene (*Marker00000819*), *UFGT* (*Marker 00029333* and *novel.8170*) in JFC and BSH compared to LFC, which may explain the red color leaves in JFC and the green color leaves in LFC. Difference in gene sequence and promoter regions of these structural genes may be the primary factors contributing to difference in gene function and expression, ultimately impacting anthocyanin biosynthesis in different clones.

The MBW protein complex was proven to activate the expression of structural genes involved in anthocyanin production. In addition to the MBW complex, other TFs such as bZIP [32,43], NAC [44], WRKY [45,46], MADS-box [47,48], and zinc finger [49,50] proteins have also shown to regulate anthocyanin biosynthesis in plants. In *Toona*, we found that MADS-box, MYB, and C2H2 possessed top five TFs related to anthocyanin derivates according to the joint analysis between transcriptome and metabolome. The gene-gene regulatory network also showed that the *SOC1*, *CPC*, and *bHLH162* were the major TFs regulating the anthocyanin biosynthesis structural genes. *SOC1* encodes a MADS box transcription factor and is involved in the regulation of flowering in response to temperature or light in various plants species, such as *Arabidopsis* [51], *Oryza sativa* L. [52], *Gossypium hissytum* [53]. In *Toona sinensis*, the young leaf color is highly sensitive to temperature, indicating a critical role of *SOC1* in temperature or light-induced anthocyanin biosynthesis. These transcription factors might regulate anthocyanin biosynthesis process by directly active or repress gene expression by binding to the promoter regions, or indirectly though protein-protein interactions in model plants and other fruits.

Plant hormones can affect anthocyanin biosynthesis through various mechanisms. Previous studies have reported that overexpression of *MdIAA26* in apple calli and *Arabidopsis* promotes the accumulation of anthocyanin, while auxin inhibit it by degrading the MdIAA26 protein [54]. Exogenous ethylene treatment increased anthocyanin accumulation in grape skins and induced the expression of structural genes (*VvPAL*, *Vv4CH*, *VvCHS*, *VvCHI*, *VvF3H*, and *VvUFGT*) and regulatory genes (*VvMYBA1, VvMYBA2,* and *VvMYBA3*) related to anthocyanin biosynthesis [55]. Our study identified nine hormone-related genes with a high correlation to regulate anthocyanins biosynthesis, which are highly expressed in JFC and play a significant role in the controlling leaf color variation in *Toona*. Leaf color is an important trait that is susceptible to both endogenous and exogenous influences. Variations in hormone signal transduction gene expression reflect the essential mechanisms by which plant hormones affect the anthocyanin biosynthesis pathway. These findings enhance our understanding of how anthocyanin biosynthesis works in *Toona* and how it can be controlled, with implications for breeding new *Toona* cultivars. Identifying and understanding the beneficial metabolic constituents and regulatory networks of *Toona* can facilitate the development and utilization of anthocyanin-enriched varieties.

## 4. Materials and Methods

### 4.1. Plant Materials

The study selected five Chinese toon clones, JinFuChun (JFC), BaShanHong (BSH), LvFuChun (LFC), FenHongXinNiang (FHXN), and HeBeiGu (HBG), which cultivated in Research Institution of Subtropical Forestry, Chinese Academy of Forestry. The colors of these varieties are: brilliant-red buds (JFC), pink buds (FHXN), light-red with a little green buds (BSH), dark-purple buds (HBG), and green buds (LFC). Fresh buds from five different clones were collected, with three replicates taken for each clone, each replicate sourced from different trees. Buds measuring 10–15 cm in length and weighing approximately 5–10 g were collected for each sample and subsequently underwent freeze-drying for RNA-seq and metabolites measures.

### 4.2. Sample Preparation and Extraction

The sample underwent freeze-drying, followed by grinding into a powder (30 Hz, 1.5 min), and subsequently, it was stored at −80 °C until required. Subsequently, 50 mg of the powder was precisely weighed and subjected to extraction using 0.5 mL of methanol/water/hydrochloric acid (500:500:1, *v*/*v*/*v*). The resulting extract was vigorously vortexed for 5 min, followed by ultrasonication for another 5 min, and then centrifuged at 12,000× *g* at 4 °C for 3 min. The residue underwent a repeat extraction using the same procedure and conditions. The supernatants obtained were collected and filtered through a 0.22 μm membrane filter (Anpel) prior to LC-MS/MS analysis.

### 4.3. UPLC Conditions

The sample extracts underwent analysis utilizing a UPLC-ESI-MS/MS system, comprising the UPLC (ExionLC™ AD, available at https://sciex.com.cn/, accessed on 17 June 2022) and the MS (Applied Biosystems 6500 Triple Quadrupole (Waltham, MA, USA), also accessible at https://sciex.com.cn/, accessed on 17 June 2022). The analytical parameters were configured as follows: UPLC:Column: Waters ACQUITY BEH C18 (1.7 µm, 2.1 mm × 100 mm); Solvent System: Water (0.1% formic acid) and Methanol (0.1% formic acid); Gradient Program: 95:5 *v*/*v* at 0 min, 50:50 *v*/*v* at 6 min, 5:95 *v*/*v* at 12 min (maintained for 2 min), 95:5 *v*/*v* at 14 min (maintained for 2 min); Flow Rate: 0.35 mL/min; Temperature: 40 °C; Injection Volume: 2 μL.

### 4.4. ESI-MS/MS Conditions

Linear ion trap (LIT) and triple quadrupole (QQQ) scans were obtained using a triple quadrupole-linear ion trap mass spectrometer, the AB Scoex Qtrap^®^ 6500+ LC-MS/MS System (Sciex, Framingham, MA, USA), equipped with an ESI Turbo Ion-Spray interface. The instrument operated in positive ion mode and was controlled by Analyst 1.6.3 software (Sciex). The ESI source operated with the following parameters: Ion Source: ESI+, Source Temperature: 550 °C, Ion Spray Voltage (IS): 5500 V, Curtain Gas (CUR): Set at 35 psi. Anthocyanins were subjected to analysis using scheduled multiple reaction monitoring (MRM). Data acquisition was conducted through Analyst 1.6.3 software (Sciex). Quantification of all metabolites was performed using Multiquant 3.0.3 software (Sciex). Further optimization of declustering potentials (DP) and collision energies (CE) for individual MRM transitions was carried out. A specific set of MRM transitions was monitored for each period, corresponding to the elution of metabolites during that timeframe.

### 4.5. Identification of Differentially Accumulated Metabolites

Significantly regulated metabolites between groups using the identification criterion of the absolute |log_2_(fold-change)| ≥ 1 and *p* value < 0.05, based on the Student’s *t*-test. Identified metabolites using the Kyoto Encyclopedia of Gene and Genomes (KEGG) compound database available at http://www.kegg.jp/kegg/compound/, accessed on 20 June 2022. Next, we mapped the annotated metabolites to KEGG Pathway database found at http://www.kegg.jp/kegg/pathway.html, accessed on 20 June 2022. The pathways that significantly regulated metabolites mapped to were integrated into MSEA (metabolite sets enrichment analysis) and evaluated for significance by checking the hypergeometric test’s *p*-values.

### 4.6. RNA Sequencing

Total RNA was extracted from frozen leaves utilizing the RNAprep Pure Plant Kit (Tiangen Biotech, Beijing, China). The Agilent 2100 Bioanalyzer (Agilent Technologies, Santa Clara, CA, USA) was used to evalute the quality of the gathered RNAs by examining their integrity. Subsequently, Poly (A) mRNA was enriched from total RNA using Oligo (dT) magnetic beads. To facilitate sequencing, Poly (A) mRNA was fragmented by an RNA fragmentation kit (Ambion, Austin, TX, USA). First-strand cDNA was produced via transcriptase reaction with random hexamer primers. Next, DNA polymerase I and RNase H enzymes were employed to create the second-strand cDNA (Invitrogen, Carlsbad, CA, USA). Following this, DNA fragments of suitable lengths were obtained, end-repaired, poly(A)-tailed, and connected with sequencing adaptors. Eventually, these fragments underwent Illumina HiSeq™ 2500 platform sequencing.

### 4.7. Transcript Profiles and Annotation

High-quality reads were obtained by processing the raw reads in fastq format using Perl scripts developed in-house. Clean reads were obtained from the raw data by eliminating adaptor sequences, low-quality reads, and reads containing polyN. All downstream analyses were based on clean, highquality data. Gene function was annotated employing several databases: KEGG pathway database, the NCBI non-redundant (Nr) database, the Swiss-Prot protein database, the euKaryotic Clusters of Orthologous Groups (KOG) database, the Gene Ontology (GO) database, and the Pfam database. We analyzed the differentially expressed genes of both groups by utilizing the DESeq R package (version 1.10.1). The DESeq R package utilizes a statistical model based on the negative binomial distribution to identify differentially expressed genes. We corrected the outcomes of all statistical tests by employing the false discovery rate of Benjamini and Hochberg to account for multiple testing. According to DESeq, genes were considered substantially differentially expressed if their adjusted *p*-value was less than 0.05. We used the top GO R package v.1, which is based on the Kolmogorov-Smirnov test, to carry out GO enrichment analysis of the differentially expressed genes. We performed pathway analysis utilizing the KEGG database (http://www.genome.jp/kegg/, accessed on 10 June 2022) to investigate relevant pathways of substantially differentially expressed genes [56,57,58].

### 4.8. Correlation Analysis of Transcriptome and Metabolome

To integrate transcriptome and metabolome datasets, Pearson correlation coefficients were utilized. Gene-metabolite coefficients were obtained by calculating the average expression levels of transcripts and metabolite contents. The fold changes for both transcriptome and metabolome data in each group were also computed. A correlation was considered significant if the Pearson correlation coefficients exceed 0.8 and *p*-values were less than 0.05 (Table 1). Significant positive correlations between transcription factors (TFs) and anthocyanin derivatives in groups JFC and LFC were detected and visually presented utilizing Cytoscape 3.10.0.

### 4.9. Verification of RNA-Seq Data by qRT-PCR

Total RNA was extracted from *Toona sinesis* leaves and reverse-transcribed using the Quantscript Reverse Transcriptase Kit. Each clone had three biological replicates, and each sample had three technical replicates. The obtained cDNA served as a template for determining gene expression levels, employing specific primers for genes linked to anthocyanin biosynthesis as well as *ACTIN* gene (used as an internal control). The primers used in the qRT-PCR analysis were shown in Table 2. The melt curve of *ACTIN* in the qPCR products was presented in Appendix A. The reaction system contained 5 μL 2 × Q3 SYBR qPCR Master Mix-Universal (TOLOBIO), 1 μL cDNA template, 0.5 μL of each forward and reverse primer, and 3 μL of RNase-free water. qRT-PCR was performed using Applied Biosystems 7500 Fast Real-Time PCR System.

## 5. Conclusions

In this study, an integrative analysis of the transcriptome and metabolome of five distinct *Toona sinensis* clones were performed to explore the related genes and metabolites associated with the anthocyanin biosynthesis. Furthermore, our analyses found that the red leaves ‘JFC’ contained the highest content of stable cyanidin, delphinidin, and peonidin, especially Cyanidin-3-O-arabinoside. The integrated analysis also identified the major transcription factor SOC1, CPC, and bHLH162. Moreover, the regulatory network construction of TFs, metabolic, structural genes reveals the underlying mechanisms of the anthocyanin biosynthesis pathway in *Toona sinensis*. Overall, this study provides a foundation for future research aimed at manipulating anthocyanin biosynthesis to improve plant coloration or to derive human health benefits.

## Figures and Tables

**Figure 1 ijms-24-15459-f001:**
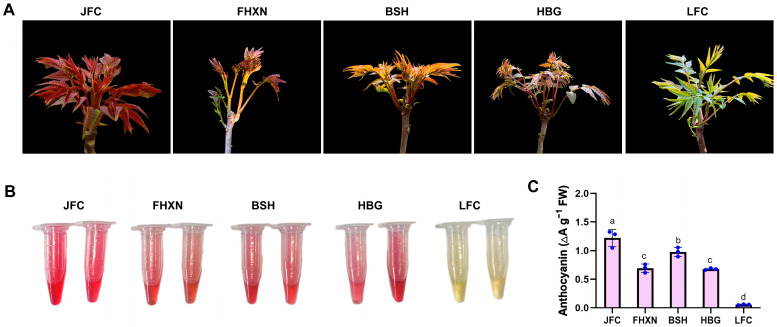
Morphological observation and anthocyanin content of leaves from different *Toona* clones (JFC, FHXN, BSH, HBG, and LYC). (**A**) The morphological characteristics of leaves from the five *Toona* clones. (**B**) The color of anthocyanin present in the leaves of these clones. (**C**) The quantification of anthocyanin content in each of the *Toona* clones. Data are shown as mean ± standard deviation, *n* = 3. Bars with different letters are significantly different at *p*  <  0.05.

**Figure 2 ijms-24-15459-f002:**
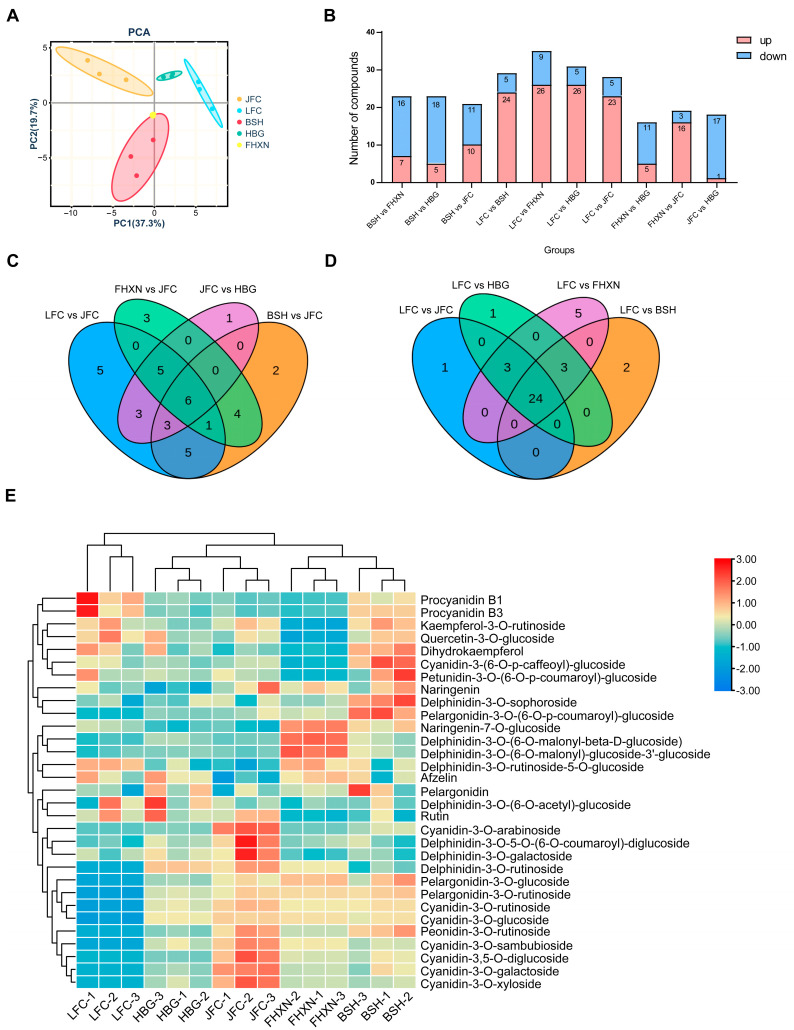
Targeted metabolome profiles of the anthocyanin biosynthesis pathway. (**A**) A PCA score plot demonstrates variations among different colored *Toona* clones. (**B**) The number of up- and down-regulated metabolites varies across different comparison groups. (**C**,**D**) Venn diagram illustrate the number of differential metabolites when compared with JFC and LFC. (**E**) Heatmaps reveals the differential metabolites present in diverse *Toona* clones. Three independent replicates of each clone are displayed in the heatmap.

**Figure 3 ijms-24-15459-f003:**
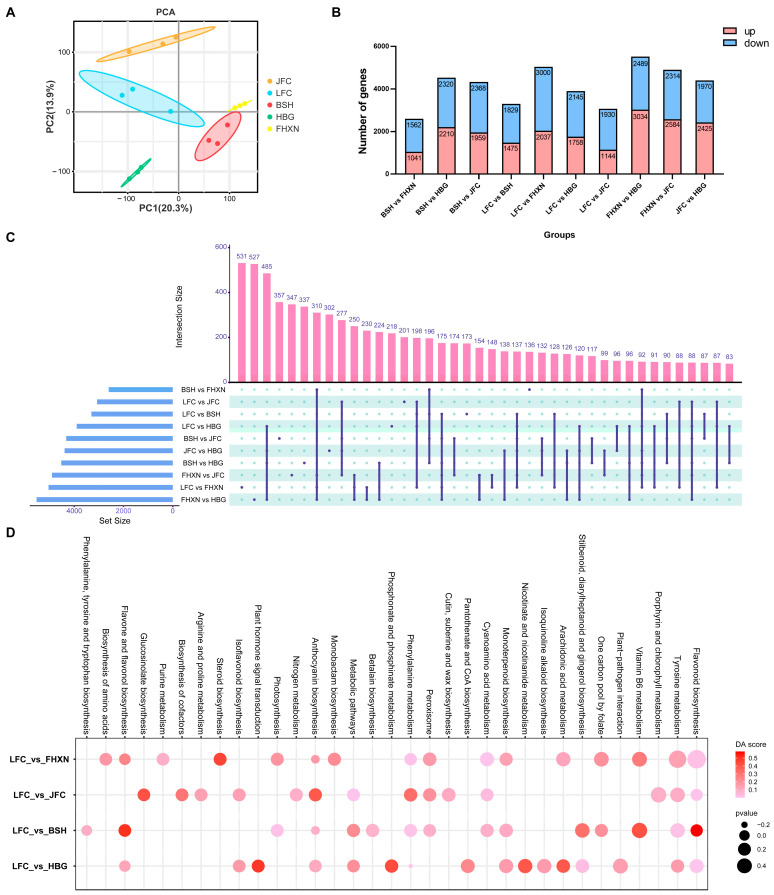
Transcriptome analysis of leaves among *Toona* clones. (**A**) A PCA score plot of the RNA-seq results of various colored *Toona*. (**B**) The number of up- and down-regulated genes differs across different comparison groups. (**C**) A shared number of DEGs across varying comparison groups. (**D**) A KEGG pathway enrichment bubble plot for DEGs between different comparison groups.

**Figure 4 ijms-24-15459-f004:**
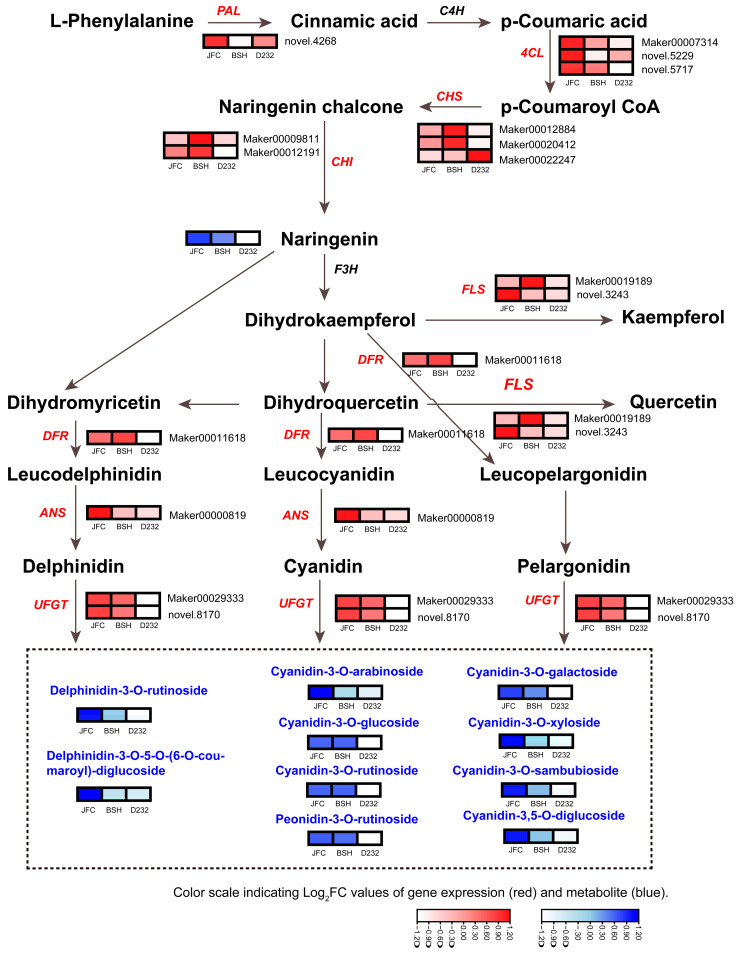
Biosynthetic pathway of anthocyanins. The construction of this pathway is based on the KEGG pathway and pertinent literature references. The red box indicated the expression of genes, and the blue box indicated the content of compounds in this pathway.

**Figure 5 ijms-24-15459-f005:**
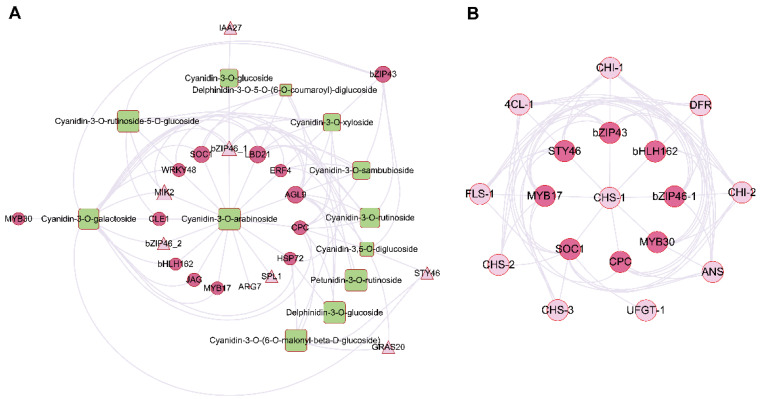
Regulatory network related to the anthocyanin biosynthesis pathway. (**A**) Connection network between regulatory genes and metabolites involved in anthocyanin biosynthesis. Purple circles represent transcription factors, light-purple triangle represent plant hormones related genes, green round rectangle represent the metabolites. The size of the cells represented the log2FC (gene or compounds). (**B**) Regulatory network between transcription factors and structural genes responsible for anthocyanin biosynthesis. Purple circles represent transcription factors, light-purple circles represent structural genes involved in the biosynthesis of anthocyanin pathway.

**Figure 6 ijms-24-15459-f006:**
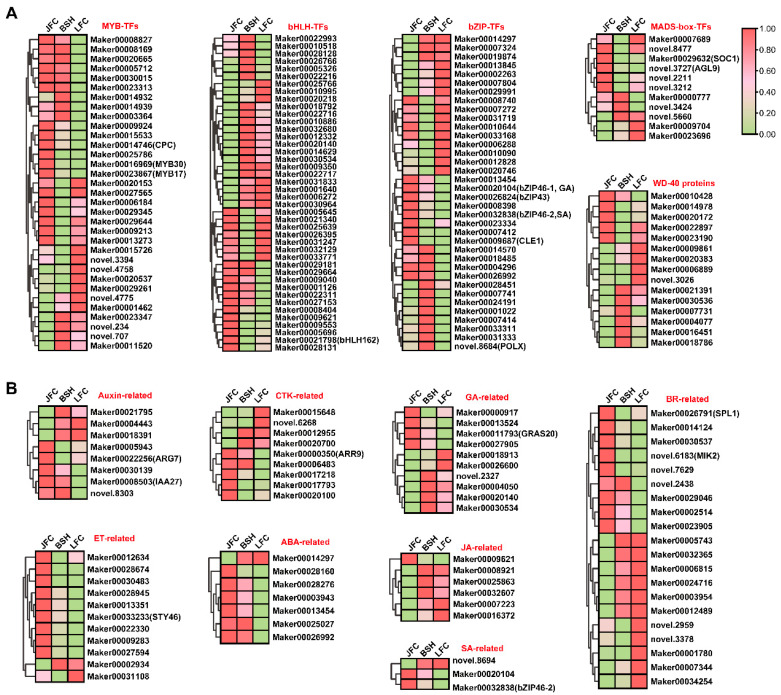
Analysis of potential transcriptional regulators linked to anthocyanins in JFC, BSH, and LFC. (**A**) The heatmap representation of RNA expression levels for various transcription factors that participated in anthocyanins production. These TFs included MYB, bHLH, bZIP, MADS-box, and WD-40 repeats. (**B**) The RNA expression level of genes responsible for plant hormones involved in anthocyanins accumulation. These genes comprise auxin, cytokinin (CTK), gibberellin (GA), brassinolide (BR), ethylene (ET), abscisic acid (ABA), jasmonic acid (JA), and salicylic acid (SA).

**Figure 7 ijms-24-15459-f007:**
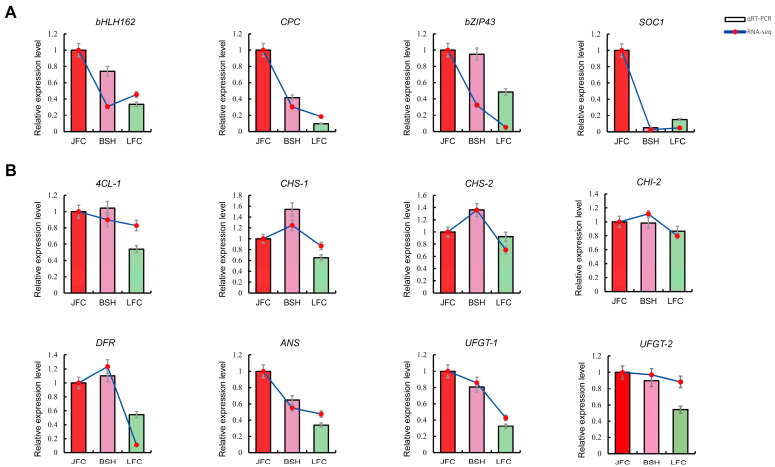
Quantitative real-time RT-PCR (qRT-PCR) and RNA-seq analysis of genes involved in anthocyanin biosynthesis pathway and putative transcription regulators. (**A**) Relative expression level of transcription factors. (**B**) Relative expression level of structural genes. Data are shown as mean ± standard deviation of three biological replicates.

**Table 1 ijms-24-15459-t001:** Correlation analysis of transcription factors and plant hormone-related genes with anthocyanin-related metabolites.

Gene ID	Log2FC_gene (JFC: LFC)	Gene Name	AGI	Gene Annotation	Compounds	Log2FC_meta (JFC: LFC)	Coefficient	*p* Value
Maker00012886	1.5011	ERF4	At5g44210	AP2 domain	Cyanidin-3-O-galactoside	7.9568	0.8479	6.52 × 10^−5^
					Cyanidin-3-O-sambubioside	5.4217	0.8049	0.0002953
					Cyanidin-3-O-arabinoside	9.025	0.8034	0.000309044
					Cyanidin-3-O-xyloside	5.2648	0.8327	0.000116859
					Delphinidin-3-O-glucoside	9.025	0.8197	0.00018361
Maker00009687	2.2469	CLE1	At2g22850	bZIP transcription factor	Cyanidin-3-O-rutinoside-5-O-glucoside	9.025	0.8066	0.000279669
					Cyanidin-3-O-galactoside	7.9568	0.8819	1.37 × 10^−5^
					Cyanidin-3-O-arabinoside	9.025	0.8696	2.54 × 10^−5^
Maker00020104	2.8632	bZIP46_1	At1g68640	bZIP transcription factor	Cyanidin-3-O-rutinoside-5-O-glucoside	9.025	0.8042	0.000301857
					Cyanidin-3-O-rutinoside	7.2774	0.8418	8.30 × 10^−5^
					Cyanidin-3-O-galactoside	7.9568	0.8692	2.59 × 10^−5^
					Cyanidin-3-O-sambubioside	5.4217	0.8169	0.000201339
					Cyanidin-3-O-glucoside	5.7951	0.8047	0.000296932
					Cyanidin-3-O-arabinoside	9.025	0.8256	0.000149917
					Cyanidin-3,5-O-diglucoside	2.6736	0.8462	6.98 × 10^−5^
					Cyanidin-3-O-xyloside	5.2648	0.8548	4.91 × 10^−5^
Maker00026824	4.2921	bZIP43	At2g22850	bZIP transcription factor	Cyanidin-3-O-rutinoside-5-O-glucoside	9.025	0.844	7.62 × 10^−5^
					Cyanidin-3-O-rutinoside	7.2774	0.8201	0.00018127
					Cyanidin-3-O-galactoside	7.9568	0.8575	4.39 × 10^−5^
					Cyanidin-3-O-sambubioside	5.4217	0.8221	0.000169159
					Cyanidin-3,5-O-diglucoside	2.6736	0.8355	0.000105219
					Cyanidin-3-O-xyloside	5.2648	0.8607	3.81 × 10^−5^
Maker00032838	1.3156	bZIP46_2	At1g68640	bZIP transcription factor	Cyanidin-3-O-galactoside	7.9568	0.8368	0.00010049
					Cyanidin-3-O-arabinoside	9.025	0.8284	0.000136002
novel.8684	2.8522	POLX	At3g45520	bZIP transcription factor	Pelargonidin-3-O-(6-O-p-coumaroyl)-glucoside	4.1336	0.8159	0.000208096
Maker00014746	2.4849	CPC	At2g46410	Myb-like DNA-binding domain	Cyanidin-3-O-(6-O-malonyl-beta-D-glucoside)	9.025	0.801	0.000332643
					Cyanidin-3-O-rutinoside	7.2774	0.8151	0.000213466
					Cyanidin-3-O-galactoside	7.9568	0.8915	8.09 × 10^−6^
					Cyanidin-3-O-sambubioside	5.4217	0.811	0.000243476
					Cyanidin-3-O-arabinoside	9.025	0.9149	1.78 × 10^−6^
					Cyanidin-3,5-O-diglucoside	2.6736	0.8358	0.000104028
					Cyanidin-3-O-xyloside	5.2648	0.8778	1.69 × 10^−5^
					Delphinidin-3-O-glucoside	9.025	0.8114	0.000240963
					Delphinidin-3-O-5-O-(6-O-coumaroyl)-diglucoside	1.0838	0.8249	0.000153878
Maker00016969	1.3602	MYB30	At4g09460	Myb-like DNA-binding domain	Cyanidin-3-O-galactoside	7.9568	0.8189	0.000188654
Maker00023867	1.6363	MYB17	At3g61250	Myb-like DNA-binding domain	Cyanidin-3-O-galactoside	7.9568	0.8493	6.17 × 10^−5^
					Cyanidin-3-O-arabinoside	9.025	0.8159	0.000207999
Maker00029632	4.4878	SOC1	At2g45660	MADS-box transcription factor	Cyanidin-3-O-galactoside	7.9568	0.9012	4.53 × 10^−6^
					Cyanidin-3-O-arabinoside	9.025	0.9525	4.40 × 10^−8^
					Cyanidin-3-O-xyloside	5.2648	0.8064	0.000281454
					Delphinidin-3-O-5-O-(6-O-coumaroyl)-diglucoside	1.0838	0.8166	0.000203571
novel.3727	4.9012	AGL9	At1g24260	MADS-box transcription factor	Cyanidin-3-O-rutinoside-5-O-glucoside	9.025	0.8712	2.35 × 10^−5^
					Cyanidin-3-O-(6-O-malonyl-beta-D-glucoside)	9.025	0.8	0.000341781
					Cyanidin-3-O-rutinoside	7.2774	0.8165	0.000204
					Cyanidin-3-O-galactoside	7.9568	0.9652	5.96 × 10^−9^
					Cyanidin-3-O-sambubioside	5.4217	0.8078	0.000269708
					Cyanidin-3-O-arabinoside	9.025	0.9734	1.07 × 10^−9^
					Cyanidin-3,5-O-diglucoside	2.6736	0.8416	8.35 × 10^−5^
					Cyanidin-3-O-xyloside	5.2648	0.8968	5.93 × 10^−6^
					Delphinidin-3-O-glucoside	9.025	0.8552	4.83 × 10^−5^
					Delphinidin-3-O-5-O-(6-O-coumaroyl)-diglucoside	1.0838	0.8735	2.10 × 10^−5^
					Petunidin-3-O-rutinoside	9.025	0.8018	0.000324031
Maker00006536	1.8404	WRKY48	At5g49520	WRKY DNA-binding protein 48	Cyanidin-3-O-rutinoside-5-O-glucoside	9.025	0.832	0.000119819
					Cyanidin-3-O-galactoside	7.9568	0.8644	3.22 × 10^−5^
					Cyanidin-3-O-arabinoside	9.025	0.8435	7.77 × 10^−5^
					Cyanidin-3-O-xyloside	5.2648	0.8093	0.000257445
novel.6183	3.5074	MIK2	At4g08850	Leucine-rich repeat receptor-like protein kinase	Cyanidin-3-O-rutinoside-5-O-glucoside	9.025	0.8304	0.000126884
					Cyanidin-3-O-galactoside	7.9568	0.9136	1.95 × 10^−6^
					Cyanidin-3-O-arabinoside	9.025	0.9219	1.03 × 10^−6^
					Delphinidin-3-O-5-O-(6-O-coumaroyl)-diglucoside	1.0838	0.8338	0.000112199
Maker00021798	1.1527	bHLH162	At4g20970	Helix-loop-helix DNA-binding domain	Cyanidin-3-O-galactoside	7.9568	0.8684	2.68 × 10^−5^
					Cyanidin-3-O-arabinoside	9.025	0.8782	1.66 × 10^−5^
Maker00003242	2.1058	JAG	At1g13400	C2H2 and C2HC zinc fingers superfamily protein	Cyanidin-3-O-galactoside	7.9568	0.8668	2.90 × 10^−5^
					Cyanidin-3-O-arabinoside	9.025	0.8657	3.04 × 10^−5^
Maker00011793	1.1426	GRAS20	At3g54220	GRAS domain family	Cyanidin-3-O-(6-O-malonyl-beta-D-glucoside)	9.025	0.8327	0.000116807
					Cyanidin-3,5-O-diglucoside	2.6736	0.8072	0.000275174
novel.4188	1.6057	HSP72	At1g56410	heat shock protein 70 family protein	Cyanidin-3-O-(6-O-malonyl-beta-D-glucoside)	9.025	0.8302	0.000127565
					Cyanidin-3-O-galactoside	7.9568	0.9167	1.55 × 10^−6^
					Cyanidin-3-O-arabinoside	9.025	0.9144	1.84 × 10^−6^
					Cyanidin-3-O-xyloside	5.2648	0.807	0.00027619
					Delphinidin-3-O-glucoside	9.025	0.8147	0.000216435
novel.4220	4.8378	LBD21	At3g11090	Lateral organ boundaries domain	Cyanidin-3-O-rutinoside-5-O-glucoside	9.025	0.8909	8.38 × 10^−6^
					Cyanidin-3-O-rutinoside	7.2774	0.8734	2.11 × 10^−5^
					Cyanidin-3-O-galactoside	7.9568	0.8584	4.21 × 10^−5^
					Cyanidin-3-O-sambubioside	5.4217	0.8307	0.00012547
					Cyanidin-3-O-arabinoside	9.025	0.8884	9.68 × 10^−6^
					Cyanidin-3,5-O-diglucoside	2.6736	0.8797	1.54 × 10^−5^
					Cyanidin-3-O-xyloside	5.2648	0.8999	4.89 × 10^−6^
					Delphinidin-3-O-5-O-(6-O-coumaroyl)-diglucoside	1.0838	0.9109	2.36 × 10^−6^
					Petunidin-3-O-rutinoside	9.025	0.8911	8.31 × 10^−6^
Maker00033233	1.0573	STY46	At4g35780	Protein tyrosine kinase	Cyanidin-3-O-(6-O-malonyl-beta-D-glucoside)	9.025	0.8235	0.000161377
					Cyanidin-3-O-galactoside	7.9568	0.8026	0.00031603
					Cyanidin-3,5-O-diglucoside	2.6736	0.8017	0.000324943
Maker00000350	1.9217	ARR9	At3g57040	Response regulator receiver domain	Cyanidin-3-O-rutinoside-5-O-glucoside	9.025	0.8952	6.53 × 10^−6^
					Cyanidin-3-O-rutinoside	7.2774	0.8032	0.000310451
					Cyanidin-3-O-galactoside	7.9568	0.842	8.23 × 10^−5^
					Cyanidin-3-O-arabinoside	9.025	0.8663	2.95 × 10^−5^
					Cyanidin-3,5-O-diglucoside	2.6736	0.833	0.000115529
Maker00026791	1.2059	SPL1	At3g57920	SBP domain	Cyanidin-3-O-arabinoside	9.025	0.8222	0.000168825
Maker00008503	1.3275	IAA27	At4g29080	AUX/IAA family	Cyanidin-3-O-glucoside	5.7951	0.8337	0.000112681
Maker00022256	1.0026	ARG7	At1g75590	Auxin responsive protein	Cyanidin-3-O-arabinoside	9.025	0.8388	9.31 × 10^−5^

**Table 2 ijms-24-15459-t002:** The primer sequences used in this study.

Primer Name	Primer Sequence F (5′-3′)	Primer Sequence R (5′-3′)
RT-bHLH162 (Maker00021798)	CAGGTTAATGGGAATCGAAAAG	CCAACCCAGTTACCAAAACAAT
RT-CPC (Maker00014746)	AAACTCGTTGGAGACAGGTG	GCAAACCCTTCTCCATGTTT
RT-bZIP43 (Maker00026824)	TCGGAGACTATGATCGGAAATC	ACTTGAAACGACAGGTTTTGGT
RT-SOC1 (Maker00029632)	CAATTGAAGCACGAAATAGCAA	TGTACCGTTTAGAGGGAAGGAA
RT-4CL-1 (Maker00007314)	AGGTTTGAGGCTTCGCAGTA	GTCCCCAGTGACAACAATCC
RT-CHS-1 (Maker00012884)	TGCAATTCTTGACCAAGTCG	GATGAACAAAACGCATGCAC
RT-CHS-2 (Maker00020412)	TGGTGCAATTGATGGACACT	TGCTTCGACCAAGCTTTTCT
RT-CHI-2 (Maker00012191)	GGGTGTTGGAAGAAAATCCA	GGTCGACTCCATCACCCTTA
RT-DFR (Maker00011618)	CCATTCCTGGCTTCAAGTGT	AAGCAATTGACCCCATTCTG
RT-ANS (Maker00000819)	GCTTTCAGGCAACTGGTCAT	TGCTCGGGAATATTTCTGCT
RT-UFGT-1 (Maker00029333)	GTGGTCGAGACTGGGAGAAG	AGGACCTGCAGTCCAAAGTG
RT-UFGT-2 (novel.8170)	GTGACAAAACCCTGGAAGGA	CAATTTTGGGAAAGCTGCAT
RT-Actin	TGATTGGGATGGAAGCAGCA	GAACATGGTTGAACCGCCAC

## Data Availability

The raw sequence data reported in this paper have been deposited in the Genome Sequence Archive (Genomics, Proteomics & Bioinformatics 2021) in National Genomics Data Center (Nucleic Acids Res 2022), China National Center for Bioinformation/Beijing Institute of Genomics, Chinese Academy of Sciences (GSA: CRA011954) that are publicly accessible at https://ngdc.cncb.ac.cn/gsa, accessed on 26 July 2023.

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
