# Peer review of "Integrated Transcriptomic and Metabolomic Analysis Reveal the Underlying Mechanism of Anthocyanin Biosynthesis in *Toona sinensis* Leaves"

_ijms, 2023, doi:10.3390/ijms242015459_

Round 1
Reviewer 1 Report
Major Comments
1. Section 2.2 and 4.3 - " absolute Log2FC (fold change) was used to determine significantly metabolites between groups". This is unclear because log2FC is not a statistical method but a metric. Please mention the statistical tests used to determine DE metabolites, alongside p-values of the significant associations.
2. Section 2.3 - could the authors elaborate on the reference gene set construction? Not clear if a single sample was chosen or a larger pangenome constructed.
3. 13,295 differentially expressed genes were identified, but how many out of 40,421 assembled could be annotated using databases?
4. Section 4.6 - since Pearson's correlation checks for the similarity between the shape of two profiles, would the determined correlations hold if other metrics, e.g. Euclidean (that accounts for shape and magnitude) were used ?
5. Figure 1 suggests two replicates were used, but it is not clear how they were combined and/or how consistent are the results between them.
Minor Comments
1. Recommend revising section 2.2 - "A total of 31 anthocyanins related metabolites were common detected. "
2. Section 2.4 - Should naringenin metabolic be naringenin metabolite?
The manuscript reads well. A few sections could be improved (see minor comments).
Author Response
> Response to Reviewer 1
Q1. Section 2.2 and 4.3 - " absolute Log2FC (fold change) was used to determine significantly metabolites between groups". This is unclear because log2FC is not a statistical method but a metric. Please mention the statistical tests used to determine DE metabolites, alongside p-values of the significant associations.
Response 1: Thanks for your great advice. We are sorry for the neglection of this error. In the analysis, we used the identification criterion of the absolute |Log2FC (fold change)| ≥ 1 and P value <0.05 based on the Student’s t-test to determine significantly different metabolites between groups. The correction was made in the corresponding section.
Q2. Section 2.3 - could the authors elaborate on the reference gene set construction? Not clear if a single sample was chosen or a larger pangenome constructed.
Response 2: Thanks for your suggestion. In the manuscript, we utilized the genome of Toona sinensis var. Heiyouchun, which marked a significant was the first chromosome-level genome assembled of Toona, as our reference genome. We compared the clean reads from our samples to this reference genome and subsequently performed gene annotations. We have added the related information and reference of the reference genome in the revised manuscript.
Q3. 13,295 differentially expressed genes were identified, but how many out of 40,421 assembled could be annotated using databases?
Response 3: Thanks for your suggestion. We are sorry for the neglection of this. In the transcriptome analysis of five Toona sinensis clones, 40,421 assembled genes were identified, including 31,647 genes annotation in reference genome. In addition, 23,909, 16,532, 25,617, 15,566, 32,348, and 25,575 unigenes were annotated by KEGG, NR, SWISSPORT, THEM, GO, and KOG database. There were 14,153 unigenes annotated by all the databases. We have added these results in the R1 manuscript.
Q4. Section 4.6 - since Pearson's correlation checks for the similarity between the shape of two profiles, would the determined correlations hold if other metrics, e.g. Euclidean (that accounts for shape and magnitude) were used?
Response 4: Thanks for your suggestion. In multi-omics investigations, the Pearson correlation coefficient was usually used to conduct the relationships between two sets of data. Meanwhile, some studies have shown that the Pearson correlation coefficient and the square of the Euclidean distance can be considered equivalent (https://blog.csdn.net/sixtyfour/article/details/80354164). Based on the published research pertaining to the integrated analysis of transcriptomics and metabolomics, most studies have relied on the Pearson correlation coefficient to detect correlations due to its high reliability (Zhang et al., 2020; Li et al., 2021). Therefore, this experiment adopts this approach to screen for candidate genes highly correlated with differential metabolites.
References:
Zhang, Z., Tian, C., Zhang, Y., Li, C., Li, X., Yu, Q., ... & Feng, S. (2020). Transcriptomic and metabolomic analysis provides insights into anthocyanin and procyanidin accumulation in pear. BMC plant biology, 20(1), 1-14.
Li, P., Ruan, Z., Fei, Z., Yan, J., & Tang, G. (2021). Integrated transcriptome and metabolome analysis revealed that flavonoid biosynthesis may dominate the resistance of Zanthoxylum bungeanum against stem canker. Journal of Agricultural and Food Chemistry, 69(22), 6360-6378.
Q5. Figure 1 suggests two replicates were used, but it is not clear how they were combined and/or how consistent are the results between them.
Response 5: Thank you for your attention. In fact, we conducted tests on three replicates of each clones to assess the anthocyanin content. However, in Figure 1B, only two samples were displayed. According to your valuable suggestion, we have incorporated symbols representing each sample into Figure1C to illustrate the consistency of these results.
Q6. Recommend revising section 2.2 - "A total of 31 anthocyanins related metabolites were common detected. "
Response 6: The sentence was revised as“A total of 31 anthocyanins were identified in all the samples”.
Q7. Section 2.4 - Should naringenin metabolic be naringenin metabolite?
Response 7: The correction was made.
Reviewer 2 Report
The manuscript is very well important and research outcomes are very interesting in promoting the use of Chinese Toon.
Detailed information on the stage and the amount of tissues used for sampling would be beneficial to readers.
The manuscript needs improvement in terms of grammar and sentence clarity.

Minor improvement needed
Author Response
> Response to Reviewer 2
Q1. Detailed information on the stage and the amount of tissues used for sampling would be beneficial to readers.
Response 1: Thanks for your suggestion. We have incorporated additional details for the sampling stage and amount in Section 4.1.
Q2. The manuscript needs improvement in terms of grammar and sentence clarity.
Response 2: Thanks for your suggestion. The correction was made.
Reviewer 3 Report
Dear Authors
The current manuscript entitled “Integrated transcriptomic and metabolomic analysis reveal the underlying mechanism of anthocyanin biosynthesis in Toona sinensis leaves” discuss the underlying mechanism of anthocyanin biosynthesis in this important tree species of China. Results suggested a strong foundation for future research aimed at manipulating anthocyanin biosynthesis to improve plant coloration or to derive human health benefits. The manuscript is very well written and presented.
The abstract and Introduction is very well organized with key informations. Results were presented in details with nice figures and data while discussed very precisely. Concluding remarks are also meeting the key message of the study.
I have some suggestions to improve the materials and method section. The plant material selection was performed among different tress or 3 replicates were chosen from the same tree?
Second, the Metabolome analysis using UPLC-MS/MS have been outsourced by Wuhan MetWare Biotechnology Co., Ltd.? There should be more details regarding this method, for example how the samples were prepared? How the UPLC was performed, the standards and quantification methods? These details are very important for the readers as they should be able to reproduce the methodology in their experiments.
Verification of RNA-Seq data by qRT-PCR also need more details. There is only one reference gene in the experiments? The number of technical and biological replicates. If only reference gene was used, please include the melt curve data for the experiments.
Thank you
Regards
Author Response
> Response to Reviewer 3
Q1. The plant material selection was performed among different tress or 3 replicates were chosen from the same tree?
Response 1: Thanks for your suggestion. Three biological replicates were conducted for the selection of plant material in each clone, sourced from distince trees.
Q2. The Metabolome analysis using UPLC-MS/MS have been outsourced by Wuhan MetWare Biotechnology Co., Ltd.? There should be more details regarding this method, for example how the samples were prepared? How the UPLC was performed, the standards and quantification methods? These details are very important for the readers as they should be able to reproduce the methodology in their experiments.
Response 2: Thanks for your suggestion. We have added the methodological details related to Metabolome analysis in Section 4.2 to 4.4 of the Materials and Methods.
Q3. Verification of RNA-Seq data by qRT-PCR also need more details. There is only one reference gene in the experiments? The number of technical and biological replicates. If only reference gene was used, please include the melt curve data for the experiments.
Response 3: Thanks for your suggestion. We have incorporated additional details regarding qRT-PCR procedures into Section 4.9 of the Materials and Methods. In the preliminary experiment, we assessed the amplification efficiency and melt curve using three reference genes, including ACTIN, UBC5B, and TUB-β. The results indicated that ACTIN performed as the most suitable reference gene. The melt curves from qRT-PCR of ACTIN gene in JFC, BSH, and LFC were present in Supplementary Figure S3.
Round 2
Reviewer 3 Report
Dear Authors
Thank you for providing the revised version of present manuscript and answering/incorporating all the queries and suggestions.
The manuscript has been significantly improved in my opinion and i do not have any further suggestions regarding this manuscript.
Regards